# "Ask the way from those who have walked it before"—Grandmothers' roles in health-related decision making and HIV pre-exposure prophylaxis use among pregnant and breastfeeding women in sub-Saharan Africa

Krishnaveni Reddy[1]*, Doreen Kemigisha[2], Miria Chitukuta[3], Sufia Dadabhai[4], Florence Mathebula[1], Siyanda Tenza[1], Thesla Palanee-Phillips[1], Julia Ryan[5], Nicole Macagna[6], Petina Musara[3], Ariane van der Straten[7,8]

1 Wits Reproductive Health and HIV Institute, School of Public Health, University of the Witwatersrand, Johannesburg, South Africa, 2 Makerere University - Johns Hopkins University Research Collaboration, Kampala, Uganda, 3 University of Zimbabwe College of Health Sciences Clinical Trials Research Centre, Harare, Zimbabwe, 4 Department of Epidemiology, Johns Hopkins Bloomberg School of Public Health Blantyre, Blantyre, Malawi, 5 Women's Global Health Imperative, RTI International, Berkeley, California, United States of America, 6 FHI 360, Durham, North Carolina, United States of America, 7 Center for AIDS Prevention Studies, University of California San Francisco, San Francisco, California, United States of America, 8 ASTRA consulting, Kensington, California, United States of America

* kreddy@wrhi.ac.za

## Abstract

HIV acquisition among pregnant and breastfeeding women in sub-Saharan Africa and vertical transmission rates remain high despite established strategies for HIV prevention. During the MTN-041/MAMMA study, we explored the influence of grandmothers (mothers and mothers-in-law of pregnant and breastfeeding women) in eastern and southern Africa on the health-related decisions of pregnant and breastfeeding women and their potential to support use of HIV prevention products. To do this we used structured questionnaires and focus group discussions with three stakeholder groups: 1) grandmothers, 2) HIV-uninfected currently or recently pregnant or breastfeeding women and 3) male partners of currently or recently pregnant or breastfeeding women. A total of 23 focus group discussions comprising 68 grandmothers, 65 pregnant or breastfeeding women and 63 male partners were completed across four study sites. Grandmothers were described as important sources of information during pregnancy and breastfeeding playing both supportive and influencer roles due to personal maternal experience and generational knowledge. While pregnant and breastfeeding women were not keen to involve grandmothers in HIV prevention decision making, they were accepting of grandmothers' involvement in a supportive role. Grandmothers expressed willingness to support pre-exposure prophylaxis use and agreed with the other two stakeholder groups that this decision should be made by women themselves or together with partners. These novel data indicate potential for grandmothers' health related supportive roles to be extended to support decision-making and adherence to biomedical

**Data Availability Statement:** All relevant data are within the paper.

**Funding:** The MAMMA study was designed and implemented by the Microbicide Trials Network (MTN). The MTN is funded by the National Institute of Allergy and Infectious Diseases (UM1AI068633, UM1AI068615, UM1AI106707), with co-funding from the Eunice Kennedy Shriver National Institute of Child Health and Human Development and the National Institute of Mental Health, all components of the U.S. National Institutes of Health. The content is solely the responsibility of the authors and does not necessarily represent the official views of the National Institutes of Health. The funders played no role in the study design, data collection and analysis, decision to publish, or preparation of the manuscript.

**Competing interests:** The authors have declared that no competing interests exist.

HIV prevention options, and possibly contribute to the decline in HIV acquisition among pregnant and breastfeeding women in these communities.

## Introduction

HIV incidence rates in pregnant and postpartum sub-Saharan African women are unacceptably high. A meta-analysis reported 4.7 cases per 100 person-years during pregnancy and 2.9 cases per 100 person-years postpartum, both of which were similar or higher than HIV incidence among female sex workers and HIV-serodiscordant couples [1]. Potential mechanisms for increased HIV susceptibility during these maternal periods include both biological and behavioral factors such as hormonal changes that affect the genital tract mucosal surfaces or immune responses [2], high rates of asymptomatic sexually transmitted infections [3] and partner HIV infection through multiple concomitant partnerships [4]. Despite established prevention strategies like provision of condoms, partner testing, male circumcision, early initiation of antiretroviral therapy, harm reduction services for women who inject drugs and management of sexually transmitted infections [5], HIV acquisition during these maternal periods and subsequent vertical transmission remain on the rise [6, 7]. Many available HIV prevention strategies rely on partner acceptance and participation; however, the ongoing HIV incidence highlights the necessity for female-initiated prevention options.

Daily oral pre-exposure prophylaxis (PrEP) consisting of tenofovir disoproxil fumarate and emtricitabine in a single pill, fixed-dose combination known as Truvada™, has been available in parts of sub-Saharan Africa (SSA) since 2016 with rollout targeting prioritized populations (i.e., sex workers, men who have sex with men, serodiscordant couples, adolescent girls and young women). Per World Health Organization guidelines, PrEP use may continue during pregnancy and breastfeeding if a woman remains at substantial risk of HIV infection [5]. The dapivirine vaginal ring (referred to as the "ring" hereafter) is a new HIV prevention option for adult cisgender women at substantial risk of HIV infection [8, 9] It is made of a flexible silicone matrix polymer containing the antiretroviral dapivirine and requires only monthly replacement [10]. It has been shown to protect women from HIV-1 in two pivotal phase III trials and their subsequent open label extension studies [11–14]. The open label extensions additionally demonstrated that women can store multiple months' ring supply in their homes and replace the ring themselves [15, 16]. The ring is not currently recommended for pregnant and breastfeeding women's use as studies to determine the safety and acceptability of the ring during pregnancy and breastfeeding are still ongoing [17]; however ring use in the periconception period was not associated with adverse effects on pregnancy or infant outcomes [18].

The effectiveness of these oral and vaginal PrEP strategies are driven by user adherence [19, 20], which can be challenging [21–23], especially with daily dosage. Barriers to PrEP use include inconsistent access to PrEP services, non-disclosure to male partners, provider bias, stigma related to HIV and PrEP use, PrEP cost, individual risk perception, low PrEP awareness, lack of social support for PrEP use, side effects as well as contextual factors such as gender and culture [24–26]. As such, innovative strategies to support uptake and adherence are warranted.

Strategies to improve PrEP use to date have mainly been focused on the user or on engaging male partner(s) and the role of family support in this regard has not been extensively researched. This is likely due to the sensitive nature of HIV prevention and fear of stigma and judgement. Family support has however been determined to have a positive impact on

patients' abilities to self-manage chronic conditions by influencing their daily behavior [27] and has potential to be extended to HIV prevention particularly among pregnant and breast-feeding women. Grandmothers are family members who play a central role in the sub-Saharan African family, as sources of information, wisdom and comfort [28]. They are seen as owners of traditional knowledge and cultural history and are important figures in pregnant and breastfeeding women's lives [29], with evidence that they can positively influence maternal knowledge and support exclusive breastfeeding [30–32]. There is however no literature indicating their possible roles as influencers for use of HIV prevention options among women generally, or specifically for those pregnant and breastfeeding. We address this gap by drawing on mixed method data gathered during the multi-site MTN-041/MAMMA study in SSA [33]. We sought to explore grandmothers' views on oral PrEP and the vaginal ring, their roles in informing decision making and their potential support of the use of these products by their pregnant and breastfeeding daughters or daughters-in-law.

## Materials and methods

### Study design

MAMMA (Microbicide/PrEP Acceptability among Mothers and Male Partners in Africa) was an exploratory, mixed method study conducted between May and November 2018 at four research clinic study sites in the following settings: Blantyre (Malawi); Johannesburg (South Africa); Kampala (Uganda) and Chitungwiza (Zimbabwe). The study included the use of structured questionnaires and single-sex focus group discussions (FGDs) with individuals independently recruited into one of three stakeholder groups: 1) Grandmothers (mothers and mothers-in-law whose daughters or daughters-in-law were currently or recently (within the previous two years) pregnant or breastfeeding, 2) HIV uninfected currently or recently pregnant or breastfeeding women aged ≥18–40) and 3) Male partners (Men aged ≥18 whose partners were recently or currently pregnant or breastfeeding). Detailed methods and primary results for this study have been previously published [33].

### Study population and settings

Participants were recruited from urban and peri-urban community settings, including street outreach, outreach at construction sites (men only), at antenatal and postnatal clinics (women only), as well as through word of mouth and community advisory board member referral. Malawian participants were recruited within Blantyre District, the country's centre of commerce and industry. South African participants were recruited within Hillbrow, an inner-city residential neighbourhood of Johannesburg that serves as a port of entry for migrants and immigrants from the townships, rural areas and the rest of Africa and as such nurtures a highly transient population. Ugandan participants were recruited within Kampala, the capital city and main centre of cash flow for Uganda's economy. Zimbabwean participants were recruited from Chitungwiza, a dormitory town to the south of Harare, and the peri-urban settlements surrounding it.

### Procedures

Data was collected at the four research clinic study sites. All participants provided written informed consent before demographic information was individually collected by site staff through the use of structured questionnaires in local languages (Chichewa in Blantyre, Malawi; isiZulu or English in Johannesburg, South Africa; Luganda in Kampala, Uganda and Shona in Chitungwiza, Zimbabwe). A staff administered behavioural assessment via structured

questionnaire was also completed with the pregnant and breastfeeding women and male participants. Gender-matched trained local social scientists fluent in local languages then facilitated the FGDs using semi-structured guides. A second trained staff member was present to assist with note taking. The FGD guides consisted of an introduction where the facilitators explained the goals and rules of the FGD followed by open-ended questions and prompts to guide the discussion. These guides were developed and pilot tested by the research team for each stakeholder group. Topics discussed included HIV risk perceptions, cultural beliefs and practices relating to pregnancy and breastfeeding, health-related decision making, key influencers and interest in two new HIV prevention products while pregnant or breastfeeding: daily oral PrEP pills and the monthly vaginal ring [11, 34]. Participants viewed a four-minute educational video (in the local language) and handled sample products immediately prior to discussing these new HIV prevention options. Participants were requested to use pseudonyms during the FGDs to protect their identities. FGDs lasted ~ two and a half hours (Minimum one hour and maximum three and a half hours) and were audio recorded, translated and transcribed in English as applicable. Facilitators completed a debriefing summary report after each FGD for rapid thematic analysis.

## Analysis

Demographic and behavioural data are presented descriptively by country. Fisher's exact tests were used to calculate differences by country with regard to women's responses on who has the most influence on their decisions during pregnancy and while breastfeeding besides themselves. For the qualitative data, analysis workshops were held for all site staff involved in qualitative data collection to conduct a preliminary analysis of the data; workshops directly informed the iterative development of the codebook used to systematically analyse all qualitative data. The codebook for this study followed a socio-ecological framework that was adapted to include the spheres of influences on future use of HIV PrEP during pregnancy and breastfeeding. This included the mother and baby dyad and the male partner or father of the baby, followed by family members (mostly grandmother of the baby, siblings and other family members), institutional and socio- structural factors [33]. FGD transcripts were coded by four data analysts using Dedoose software (v7.0.23). An acceptable level of intercoder reliability was set and maintained at approximately 80% agreement. The analysis team met weekly to discuss coding questions, issues, emerging themes and data saturation as well as to resolve discrepancies. Coded data reports were further summarized thematically into analytical memos that were reviewed by site teams [33, 35]. For this analysis, we looked at family influences specifically grandmothers as they emerged as being important influencers from responses to structured questionnaires and in FGDs. Data coded for "FAMILY" were extracted from all FGD transcripts and stratified by stakeholder group type (i.e., grandmothers, pregnant or breastfeeding women and male partners) in addition to country. Additionally, a product-focused acceptability framework [36] was used to understand prospective acceptability of the two HIV prevention methods. For this, data coded for "PILL", "RING" and "PREFERENCE" were extracted from Grandmother FGD transcripts and stratified by country. Data reports were then thematically analysed by representatives of the four research clinic study sites into analytical memos that were reviewed by the writing team biweekly to discuss coding questions and emerging themes.

## Ethics statement

The study protocol was approved by the Western Institutional Review Board located in Olympia, Washington, USA as well as local institutional review boards at each of the study sites and

was overseen by the regulatory infrastructure of the U.S. National Institutes of Health and the Microbicide Trials Network (MTN). Written informed consent was obtained from all participants enrolled in the study.

## Results

In total, 196 individuals joined one of 23 FGDs (Two FGDs were conducted with each stakeholder group at each study site except for grandmothers in Blantyre, Malawi where only one FGD was conducted). Demographic data are presented descriptively by stakeholder group and country in Table 1. The mean age of grandmothers was 50 years (min 36, max 69) with most grandmothers living with their children (81%, N = 55). The mean age of pregnant and breastfeeding women and male partners was 27 years (min 19, max 40) and 31 years (min 19, max 54) respectively. The South African pregnant and breastfeeding women and male partners differed from those in the other settings with regards to marital status and living arrangements. Most were single (93% of pregnant or breastfeeding women and 92% of male partners) and majority (67% of pregnant or breastfeeding women [N = 10] and 58% of male partners [N = 7]) were living with adult family members including parents and siblings. In the other settings, most pregnant and breastfeeding women and male partners were married (83%–94%) and living with their spouse or primary partner (79%–94%).

Overall, data collected from the FGDs described grandmothers as important sources of information, playing both supportive and influencer roles, due to personal maternal experience and generational knowledge. All stakeholder groups agreed that HIV prevention related decision making should be made by pregnant and breastfeeding women themselves or together with partners. However there was indication from the pregnant and breastfeeding women group that grandmothers could be involved in HIV prevention product use in a supportive role if this was disclosed to them. Importantly, grandmothers themselves expressed willingness to support PrEP use.

### Views on influential decision-makers during pregnancy and breastfeeding

Data from the behavioural assessment (Table 2) indicated that the majority of pregnant and breastfeeding women in Malawi, Uganda and Zimbabwe, thought that besides themselves, the father of the baby had the most influence on their decisions during pregnancy (60%–88%) and while breastfeeding (53%-92%). South African women, however, reported that their mothers (40%) had more influence than the baby's father (20%–27%).

FGDs revealed similar findings to the behavioural assessment, with household composition and living arrangements appearing to impact who the key decision-making influencers were. South African grandmothers, pregnant and breastfeeding women, and male partners emphasized that it is the grandmothers (mother of the pregnant or breastfeeding woman) who make decisions, especially in cases where the pregnant or breastfeeding women live with their mothers or returned home to their mothers to give birth (even if married). Grandmothers help look after the baby and therefore have authority: *The decision we take or follow as the family are more important that those from the clinic because as a nursing mother you live, and sleep with your granny and mother in the house, and they help look after the baby, so you must listen to them anything you do. [Dineo, Grandmother, 58, South Africa]*

Decision making may also be impacted by lobola (payment a male partner or head of his family gives to the woman's family in gratitude for allowing the marriage) or damages (payment made if a woman is impregnated before marriage to show that the male partner's family accepts the baby as their own) as expressed by male partners and pregnant or breastfeeding women in South Africa as well as male partners in Zimbabwe: *I am always afraid of*

**Table 1. Demographic information across the four participating study sites.**

| | Grandmothers (N = 68) | | | | Pregnant and Breastfeeding Women (N = 65) | | | | Male Partners (N = 63) | | | |
|---|---|---|---|---|---|---|---|---|---|---|---|---|
| | Malawi (N = 10) | South Africa (N = 20) | Uganda (N = 21) | Zimbabwe (N = 17) | Malawi (N = 15) | South Africa (N = 15) | Uganda (N = 18) | Zimbabwe (N = 17) | Malawi (N = 16) | South Africa (N = 12) | Uganda (N = 19) | Zimbabwe (N = 16) |
| **Mean Age (Years)** | 50.6 (39–69) | 54.9 (36–67) | 47.1 (37–63) | 46 (36–63) | 26.7(21–34) | 28.0 (22–40) | 27.2 (19–40) | 26.6 (19–38) | 30.2(19–53) | 33.0 (27–49) | 32.4 (23–54) | 27.0 (19–45) |
| **Secondary education complete** | 1 (10%) | 6 (30%) | 6 (29%) | 6 (35%) | 6 (40%) | 11 (73%) | 4 (22%) | 12 (71%) | 6 (38%) | 8 (67%) | 9 (47%) | 12 (75%) |
| **Earning own income (#)** | 7 (70%) | 3 (15%) | 19 (91%) | 11 (65%) | 9 (60%) | 0 | 12 (71%) | 6 (35%) | 13 (81%) | 4 (33%) | 17 (90%) | 14 (88%) |
| **Religion** | | | | | | | | | | | | |
| Christian | 10 (100%) | 19 (95%) | 16 (76%) | 16 (94%) | 14 (93%) | 15 (100%) | 17 (94%) | 17 (100%) | 15 (94%) | 9 (75%) | 14 (74%) | 15 (94%) |
| Muslim | 0 | 0 | 5 (24%) | 0 | 1 (7%) | 0 | 1 (6%) | 0 | 1 (6%) | 1 (8%) | 5 (26%) | 0 |
| None | 0 | 1 (5%) | 0 | 1 (6%) | 0 | 0 | 0 | 0 | 0 | 2 (17%) | 0 | 1 (6%) |
| **Marital status** | | | | | | | | | | | | |
| Single | 0 | 12 (60%) | 2 (10%) | 0 | 0 | 14 (93%) | 1 (6%) | 0 | 1 (6%) | 11 (92%) | 3 (17%) | 0 |
| Married | 5 (50%) | 3 (15%) | 7 (33%) | 13 (76%) | 14 (93%) | 1 (7%) | 16 (89%) | 16 (94%) | 15 (94%) | 1 (8%) | 15 (83%) | 15 (94%) |
| Separated or divorced | 2 (20%) | 3 (15%) | 9 (43%) | 0 | 1 (7%) | 0 | 1 (6%) | 1 (6%) | 0 | 0 | 0 | 0 |
| Widowed | 3 (30%) | 2 (10%) | 3 (14%) | 4 (24%) | 0 | 0 | 0 | 0 | 0 | 0 | 0 | 0 |
| Other | 0 | 0 | 0 | 0 | 0 | 0 | 0 | 0 | 0 | 0 | 0 | 1 (6%) |
| **Household composition*** | | | | | | | | | | | | |
| Lives alone | 0 | 0 | 1 (5%) | 0 | 0 | 1 (7%) | 1 (6%) | 0 | 0 | 2 (17%) | 1 (5%) | 1 (6%) |
| Spouse or primary partner | 5 (50%) | 4 (20%) | 5 (24%) | 12 (71%) | 13 (87%) | 3 (20%) | 15 (83%) | 16 (94%) | 15 (94%) | 4 (33%) | 15 (79%) | 15 (94%) |
| Mother and/ or father | 2 (20%) | 3 (15%) | 1 (5%) | 1 (6%) | 1 (7%) | 5 (33%) | 1 (6%) | 1 (6%) | 0 | 7 (58%) | 1 (5%) | 2 (13%) |
| Sibling(s) | 1 (10%) | 5 (25%) | 3 (14%) | 1 (6%) | 3 (20%) | 8 (53%) | 1 (6%) | 0 | 2 (13%) | 3 (25%) | 4 (21%) | 2 (13%) |
| Grandparent (s) | 0 | 1 (5%) | 0 | 1 (3%) | 0 | 1 (7%) | 0 | 0 | 2 (13%) | 2 (17%) | 0 | 0 |
| Other relative(s) | 1 (10%) | 5 (25%) | 3 (14%) | 1 (6%) | 1 (7%) | 1 (7%) | 0 | 4 (24%) | 3 (19%) | 1 (8%) | 1 (5%) | 0 |
| Child(ren) | 8 (80%) | 16 (80%) | 18 (86%) | 13 (77%) | 11 (73%) | 8 (53%) | 11 (61%) | 14 (82%) | 13 (81%) | 1 (8%) | 11 (58%) | 10 (63%) |
| Grandchild (ren) | 6 (60%) | 13 (65%) | 10 (48%) | 2 (12%) | 1 (7%) | 0 | 0 | 0 | 0 | 0 | 0 | 0 |
| Other | 0 | 1 (5%) | 0 | 1 (6%) | 3 (20%) | 0 | 1 (6%) | 2 (12%) | 0 | 1 (8%) | 0 | 0 |
| Adult Family Member | 3 (30%) | 13 (65%) | | 4 (24%) | 4 (27%) | 10 (67%) | 1 (6%) | 5 (29%) | 7 (44%) | 7 (58%) | 6 (32%) | 4 (25%) |

* Participants could select multiple responses

*complications when you have not yet paid lobola to your in laws... You won't have any say in your relationship. [Tinashe, Male partner, 19, Zimbabwe]*

In the settings outside South Africa, most participants did not generally consider grandmothers as decision makers, with a few exceptions related to non-payment of lobola, living in close proximity and because the women may behave irrationally (e.g., having variable moods, getting upset) during pregnancy and need someone to decide on their behalf: *I would think it is that person who is near you, it can be your husband, your mother and the health worker because pregnant women sometimes behave funny. [Esther, Pregnant woman, 22, Uganda]*

**Table 2. Pregnant and breastfeeding women's responses when asked about who has the most influence on their decisions during pregnancy and while breastfeeding besides themselves.**

| Influencer* | Pregnancy (65 responses) | | | | | Breastfeeding (48 responses) ⸰ | | | | |
|---|---|---|---|---|---|---|---|---|---|---|
| | Malawi | South Africa | Uganda | Zimbabwe | Fisher's Exact *p*-value | Malawi | South Africa | Uganda | Zimbabwe | Fisher's Exact *p*-value |
| Father of baby | 9 (60%) | 4 (27%) | 11 (61%) | 15 (88%) | 0.001 | 11 (92%) | 2 (20%) | 8 (53%) | 9 (82%) | 0.007 |
| Mother | 2 (13%) | 6 (40%) | 2 (11%) | 0 | | 1 (8%) | 4 (40%) | 3 (20%) | 1 (9%) | |
| Mother-in-law | 1 (7%) | 0 | 1 (6%) | 0 | | - | - | - | - | |
| Doctor | 3 (20%) | 0 | 3 (17%) | 2 (12%) | | 0 | 0 | 4 (27%) | 0 | |
| Nurse | - | - | - | - | | 0 | 2 (20%) | 0 | 0 | |
| Other | 0 | 4^ (27%) | 0 | 0 | | 0 | **2(20%) | 0 | 1• (9%) | |
| No response | 0 | 1 (7%) | 1 (6%) | 0 | | - | - | - | - | |

* Participants were permitted to choose one option

⸰ Only 48 women were currently breastfeeding or had ever breastfed

^ Included aunts (2), sister (1) and no one else (1).

** Included aunts (1) and sister (1).

• Includes no one else (1)

Grandmothers tended to portray their daughters and daughters-in-law as naïve during maternal times, especially if it was a first pregnancy, seeing themselves in a privileged position to support and educate them due to their own maternal experiences and expertise and as custodians of knowledge passed down through generations:

*In Zulu there is a saying that say, "Ask the way from those who have walked it before", even if people don't have mothers, you can see from neighbours or you can ask from your aunt or any elder.*

*[Sindiswa, Grandmother, 36, South Africa]*

This included advising on in a range of domains including cultural practices, correct food and drink, and health seeking behaviour. Grandmothers in Malawi and South Africa said they also guided their daughters or daughters-in-law in sexual matters (see Table 3).

Overall, pregnant and breastfeeding women and male partners confirmed the value of grandmothers' traditional knowledge and advice, ascribing them with legitimacy since they had healthy pregnancies before allopathic medicine became the standard. A reliance on maternal grandmothers was emphasized in many instances: *But if you notice our mothers grew up preferring traditional doctors than medical doctors and they went through all this process without consulting [medical] doctors, of which their pregnancy and their children were healthy. [Asanda, Pregnant woman, 26, South Africa]*

Given the importance of grandmothers in providing cultural information and traditional wisdom to pregnant and breastfeeding women, we further explored grandmothers' willingness to support their pregnant or breastfeeding daughters/daughters-in-law in the use of PrEP in the future as well as their views on the ring and oral PrEP for HIV prevention, including the likelihood of cultural acceptance or resistance to these products.

## Grandmothers' willingness and motivation to support HIV prevention product use

Across all groups, the majority views about HIV testing and prevention related decision making was that it is a decision made by pregnant or breastfeeding women alone or together with

**Table 3. Grandmother advice and support provided during pregnancy or breastfeeding as described by participants during FGDs.**

| Domains | Exemplary Quote |
|---|---|
| **Grandmothers' advisory roles** | |
| Diet | *When our partners are pregnant especially for the first time, we usually go and consult our parents to know what foods a pregnant woman is supposed to eat, how you have to handle her so that the pregnancy remains well, and she is also healthy. [Emma, Male partner, 23, Uganda]* |
| Health seeking behaviour | *She treat her baby like a doll. I know more; so, I am the one who is supposed to determine that she must take the baby to the clinic or say no here you must take a baby to the clinic not to the doctor, but to the clinic. [Dineo, Grandmother, 58, South Africa]* |
| Cultural practices | *So, there are some of the things that she will go and learn concerning her health, and other cultural practices that are done so that the baby will not grow up as a feeble. I heard that our in-laws will teach their daughter what to do so that the baby will grow healthy. [Pizza, Male partner, 28, Zimbabwe]* |
| Sexual matters | *So while there is this saying that men cannot restrain themselves, there is this sexual practice called "Ukusoma," [thigh sex], a man ejaculate outside not inside the womb. And we grew up knowing that as woman you should do that type of sex after you have delivered, your parents tell you. That is why you have to deliver at your parents' place so that your mother can be open and guide you. Then after delivery you will go back to your marriage home, after three to six months. [Guest, Grandmother, 67, South Africa]* |
| Maternal Behaviour | *A mother [grandmother] because you have to tell your daughter that now that you are pregnant you should not walk about in the streets, you should not stand by the gate, you should not eat standing, such things. [Zintle, Grandmother, 63, South Africa]* |
| **Grandmothers' supportive roles** | |
| Practical | *I then go back home and in case my husband has no money to look after me I then go to my mother-in-law and she will provide anything that I might need. Yes, health workers will provide the health care and then if my husband fails to get what to eat then I go to my mother-in-law for help. [Agatha, Pregnant woman, 21, Uganda]* |
| Emotional | *Most times when you are worried when you are pregnant and the worries are coming because of the husband you are with, you go and share your worries with your mother because she is the one you are close to. [Lucy, Breastfeeding woman, 29, Malawi]* |
| Instrumental | *I think this thing depends on my ability when my daughter goes out with a boyfriend; I must check her when she leaves, call her and tell her to take her pills with [her]. I love her and concerned about her health while pregnant so I have to always ask her if she has taken her pills. [Apple, Grandmother, 54, South Africa]* |

their partners with emphasis placed on openness and tolerance for health seeking behaviours during these times: *When it comes to making decisions in a family, it is good to decide together as husband and wife. . .the most important thing is that there should be openness in the family. . .there is need to make decisions together not each one by themselves. [Davie, Male partner, 24, Malawi]*

Pregnant and breastfeeding women additionally indicated that grandmothers should be kept out of decisions to use HIV prevention products because of cultural taboos that forbid discussing sexual issues with your mother or mother-in-law, and issues of confidentiality however, they thought grandmothers could be involved in HIV prevention product use in a supportive role if this was disclosed to them: *It depends if you want to tell them because at the end of the day it is your life at risk so if you have to inform them already that you are taking either the vaginal ring or oral PrEP that I am taking this because of this. If they are interested, they will support you if they are not they won't. [Apple, Pregnant woman, 24, South Africa]*

Grandmothers themselves expressed a high level of willingness to support their daughters and daughters-in-law to use HIV prevention methods. Their main motivation to support HIV prevention product use was to prevent a disease with no cure, caused in many cases by male partners' having other sexual partners/relationships and disliking condom use, and eventually

resulting in the loss of children to HIV. Additional reasons cited included the responsibility of having to care for an HIV infected daughter or daughter-in-law and infant: *The baby may be infected with the virus and instead of caring for one person I have two people to take care of.* *[Agatha, Grandmother, 61, Uganda]*

Grandmothers indicated they could offer practical, emotional and instrumental support to ensure their daughter's or daughter-in-law's high adherence to PrEP. For oral PrEP it included daily reminders, ensuring that she takes her pills with her when she goes out or travels or providing the pills themselves to ensure she takes it daily: *I would like her to take those drugs every day because some people forget or at times they are negligent about their lives so if she lives in next to me I would like to give her the pills by myself. We may agree to a certain time that at such a time I will knock at your door so that you take it [pill]. [Esther, Grandmother, 44, Uganda]*

Support for ring use included facilitating access to the ring and ensuring it is inserted appropriately: *I would accompany her to go and change the ring and if she can't, she can send me to fetch it for her, when I get home with it I would say let's go the bedroom I have your parcel and see her inserting it. [Dineo, Grandmother, 58, South Africa]*

## Grandmothers' views on new biomedical prevention tools

Overall, grandmothers embraced the concept of new HIV prevention products for pregnant and breastfeeding women, beyond condoms: *The advantage with these products is that they contain medicines that can kill the virus unlike a condom which when worn incorrectly can burst then you get HIV. [Mbuya Shava, Grandmother, 43, Zimbabwe]*

Grandmothers expressed relief that options to protect both their daughters and their unborn grandchildren were forthcoming, although establishing the safety of these would be crucial: *I would be happy for our daughters to use it [oral PrEP pill] at least they would be protected from their partners, and their babies would be. . .Now the issue of health; I mean the risk of how their bodies would tolerate the tablet. The side effects are my concerns because the tablets must be consumed. [Mamorena, Grandmother, 58, South Africa]*

Indeed, grandmothers expressed several concerns which may impact future endorsement of these products (see Table 4). Ring concerns included its size, whether it would enlarge the vagina and be painful to the woman using it, side effects for the foetus or infant from the medication or causing newborn injuries during delivery.

*I was just thinking that if it remains inside it can hurt the baby during birth. Maybe the birth canal would be too narrow.*

*[Mbuya vaPinky, Grandmother, 48, Zimbabwe]*

Several partner-related concerns were raised as well:

*Because once he feels the ring, that is when he might want to know what's inserted inside and start accusing the wife of prostitution. When the ring is actually being helpful.*

*[Mbuya Zvakanaka, Grandmother, 63, Zimbabwe]*

Perceived negative outcomes for oral PrEP use raised by grandmothers included HIV risk related to forgetting to take the pills, HIV stigma, side effects to the mother and safety of the developing foetus, including potential for miscarriage in the first trimester (Table 4). Miscarriage concerns were related to medication bitterness in Malawi, Uganda and Zimbabwe: *Our parents used to say that a pregnant woman should not take drugs that are bitter because some*

**Table 4. Grandmother views on the ring and oral PrEP for pregnant and breastfeeding women.**

| Product related attribute/theme | Ring | Oral PrEP |
|---|---|---|
| *Dosage form/ familiarity and comfort* | *Since it is a new thing that is being introduced, when she starts using it, will she feel comfortable having it inserted, is it not painful the moment she starts using it? [Mbuya Rutendo, 44, Zimbabwe]* | *Maybe most of all have accepted that HIV is an existing condition and it affects everyone so a person who will say "HIV," no that person is behind times because all of us are HIV and taking pills. So I think pills are acceptable I don't think anyone will have a problem while pregnant. [Guest, 67, South Africa]* |
| | *Isn't it's big? I just wondered if our vaginas are big like the ring. Will the ring not enlarge the vagina? [Mbuya vaPinky, 48, Zimbabwe]* | |
| *Dosage form/ mechanism of action* | *I prefer the ring because it stays in there, it's safer, and you cannot forget it and you don't have to drink it like a pill which might have side effects for both a pregnant mother and the foetus. [Yellow, 43, South Africa]* | *I am just wondering, like if I took it yesterday and today I forget to take it, how long does it take to work in my body, if on the day I forget it I engage in sex wouldn't I be putting myself at risk? [Mamorena, 58, South Africa]* |
| *Dosage form/User burden* | *Because if she inserts the ring, she will not forget like she will when taking the pill. She might take them and sometimes forget. Or she might go for a funeral and you will hear her say that, "I forgot my pills." [Gogo Munyemba, 42, Zimbabwe]* | *I don't agree with the pill, it is the vaginal ring only. One can easily forget to take the pill. [Rose, 46, Uganda]* |
| *Discretion of use/ HIV stigma* | | *She might find it hard to take them when surrounded by people. She will fear that people will think she will be using an ARV to treat HIV, so it may be difficult for her to explain to people about the pill she is taking. [Mbuya Shava, 43, Zimbabwe]* |
| *Discretion of use/ gendered relationships* | *I think some people will use it discreetly because men are always against whatever is implemented. One can say "why don't you trust me? Do you think I have so many other sexual partners?" So some women will use it discreetly. [Mary, 54, Uganda]* | |
| *Safety/Mother or baby* | *I do not know if it does not have some side effects that can affect the baby. [Mbuya vaPinky, 48, Zimbabwe]* | *. . . the bitter medicine may interfere with the baby in the womb and lead to a miscarriage. [Zaina, 42, Uganda]* |
| *Safety/during labour* | *What happens when a pregnant woman with a ring gets into labor and wants to deliver the baby there and when the ring is still in place? [Mbuya Zvakanaka, 63, Zimbabwe]* | |
| | *The baby might react to the medicine from the ring during delivery, and it may affect her. [Gogo Tadiwa, 40, Zimbabwe]* | |
| | *I am afraid that maybe the baby might put its little head inside the ring. [Mbuya Muti, 62, Zimbabwe]* | |

*drugs can cause abortion. It all depends on how strong one's blood is. Some women have strong blood and some have weak blood.* [Evelyn, Grandmother, 47, Malawi]

When elaborating on the stigma associated with taking ARVs for HIV treatment, some added that having HIV these days was not seen as stigmatizing as it was previously, as the disease is commonplace, impacts everyone to some degree and is accepted by many. There was a general feeling that the health of their daughters or daughters-in-law should be prioritized and should override worry related to stigma: *It's high time that you don't focus on checking who is saying what about your health, it is up to you whether your health stays safe no matter who says what.* [Pinki, Grandmother, 43, South Africa]

Grandmothers acknowledged that if these products have been well researched and tested, the safety concerns they raised could be overcome with the correct information from health-care providers (HCPs) who are best positioned to guide pregnant and breastfeeding women on prevention method use: *These pills have been examined and tested first, it's not like they will just come from the blues and get imposed on us. So, teachings are important, like we are taught when we go to clinics. So, people should be taught until they come to know and accept the new products so that no one will think that they will be affected negatively by them. [Mbuya Peter, Grandmother, 45, Zimbabwe]*

The ring appeared to be favoured by grandmothers over oral PrEP given its discrete use (without partner knowledge), its monthly duration and lower use burden (less likely to forget to use) and route of administration as, per grandmothers, it avoids the need to ingest which might cause side effects for the pregnant mother and foetus: *Because once you insert the ring you spend the whole month with it without any challenges but with the daily oral PrEP people will forget. [Mbuya Tsitsi, Grandmother, 42, Zimbabwe]*

There was a general view that pregnant and breastfeeding women should be able to protect themselves and both products were referred to as a "*women's defence*" to shield infection brought into their relationship by male partners, particularly in situations where women do not have decision making power (e.g., cannot negotiate condom use): *There is a need to quietly protect yourself since it is culturally accepted that a man can just find someone to have sex with and they do it without using protection. At home wives do not use protection, so it will be good if I protect myself because if he gets infected, I will block the virus and he will have his own virus. [Mbuya Rutendo, Grandmother, 44, Zimbabwe]*

## Grandmothers' views on cultural misalignments with the ring and oral PrEP

The majority of grandmothers did not expect oral PrEP or the ring to conflict with cultural beliefs and practises around pregnancy and breastfeeding. They did not feel that the taste of oral PrEP would be culturally problematic as other biomedical interventions used during pregnancy are bitter (e.g., Fansidar for malaria) as are some traditional medicines but they are still used. One grandmother likened HIV-negative pregnant women taking oral PrEP to HIV-positive pregnant women taking ARVs, which is widely accepted: *There is no cultural belief that can prevent a pregnant woman from taking Truvada because the Government has a policy that all pregnant women who are found to be HIV-positive at the ANC, when the pregnancy is term, they are given drugs to take in order to prevent the unborn baby from contracting HIV and AIDS. So I feel this drug called Truvada is like the same as that drug [for treatment]. [Tadala, Grandmother, 40, Malawi]*

Interactions between traditional vaginal products and the ring (possibly impacting efficacy of the ring) were raised among some grandmothers but they indicated they would discourage their pregnant and breastfeeding daughters or daughters-in-law from these practices when using the ring so that they may be protected from HIV:

> *I will tell her to use the ring and do birth preparation practices that does not require the use of herbs. She will use the ring to prevent HIV and stop using herbs. I will encourage her to use the soap only because the soap does not contain any drug that can cause some side effects.*
>
> *[Mbuya Rarara, Grandmother, 36, Zimbabwe]*

Overall, grandmothers indicated that HIV prevention is the main priority, different illnesses require different approaches and, as there are no traditional medications available to

stop HIV, the ring and oral PrEP are the tools to achieve this key goal: *I don't see why there should be a clash because there's nothing that can prevent HIV except for things like these, the ring and the pills. [Yellow, Grandmother, 43, South Africa]*

## Discussion

We used mixed method data collected during the MAMMA study to explore grandmothers' roles in the health-related decision making of their pregnant and breastfeeding daughters or daughters-in-law, their views on the ring and oral PrEP for HIV prevention and their willingness to support their pregnant and breastfeeding daughters' or daughters'-in-law use of these products. Grandmothers were described as custodians of traditional and cultural practices and a source of information and support to their pregnant and breastfeeding daughters and daughters-in-law. That said, the majority of participants across stakeholder groups and study settings beside South Africa indicated that grandmothers roles in health related decision making during these maternal times was secondary to male partners. Grandmothers in South Africa however were described as having more decision-making capacity which is likely linked to the fact that the pregnant or breastfeeding woman would be living in their parents' home. With regard to actual HIV prevention product use, most participants reported that this decision should be made by women themselves or both partners together and not involve grandmothers however the potential supportive role of grandmothers was endorsed in all settings, and again more strongly so in South Africa.

Strategies to improve PrEP use among women to date have been focused on the user or on engaging male partner(s) and the role of family support has not been extensively researched. Grandmothers appeared to readily understand the purpose of the ring and oral PrEP with only a brief introduction (short video and sample product handling) and expressed a high level of willingness to support their pregnant and breastfeeding daughters and daughters-in-law in their use of these products. They expressed the need to protect their children and unborn grandchildren consequently preventing themselves from returning to a mothering role in their old age and serving as caregivers for their grandchildren orphaned by HIV, a frequent occurrence of the pandemic in SSA [37, 38]. While they did raise concerns about both HIV prevention products, grandmothers recognized that these products have been researched and tested and that these concerns as well as cultural conflicts could be overcome through education and access through HCPs, trained to counsel pregnant and breastfeeding women on their proper use. Many of their concerns about HIV prevention product use were reflective of known concerns of oral and vaginal PrEP users among multiple populations and trial settings [39, 40] and not based on clinical evidence. Both oral PrEP and the ring are safe to use. Common side effects include those of gastrointestinal (oral PrEP) or vaginal (ring) origin and resolve with time.

These data bring to light new insights and opportunities to potentially utilize grandmothers to support PrEP uptake and use particularly among pregnant and breastfeeding women. Efforts have been made to enhance the skills of grandmothers to discuss issues related to sex and sexuality with young girls to reduce unintended pregnancy [41] as well as HIV acquisition [42] and their support has been shown to be important for prevention. It is therefore possible that their supportive roles during pregnancy and breastfeeding can be similarly harnessed to promote HIV prevention product uptake and adherence among their daughters and daughters-in-law. Further consultations with grandmothers on how to best engage them, with a focus on PrEP education and adherence support mechanisms and strategies, are needed.

The MAMMA study had several limitations. Firstly, while male partners and grandmothers were recruited from various urban and peri-urban community settings, which may best reflect

overall community views, pregnant and breastfeeding women were recruited mainly from antenatal and postnatal clinics and, as such, represent views of women who have access to healthcare services. It is possible that pregnant and breastfeeding women without access to healthcare services may rely more on family in terms of their decision making. Secondly, participants were HIV prevention product naïve which meant that they had no direct experience with the products when discussing them. This was purposefully done to elicit oral PrEP and ring discussion and ascertain perspectives representative of the product naïve communities into which these products are being or will be introduced for rollout. Thirdly, it was difficult to ascertain during analysis whether some responses during the discussions were related to mothers or mothers-in-law specifically, as the terminology was interchanged with the more generic "grandmother" term. Fourthly, although facilitators ensured comfortable and casual FGD venues and discussions, responses may have been swayed by social desirability bias due to the FGDs occurring within clinic settings in order to maintain privacy and confidentiality. Fifthly, transcripts were not returned to participants for review as this would have increased the risk of loss of confidentiality, given the group setting of the FGDs. Nevertheless, study participants were invited to study dissemination stakeholder meetings and had the opportunity to hear and provide feedback on the overall study results during these meetings. Lastly, qualitative data analysis is interpretative; however, representatives from each setting were involved during analyses and biweekly meetings to discuss coding, emerging themes and to maximize consensus. While these limitations may have impacted our analysis to some extent, the multi-country sample and discussions that were mostly aligned across the different settings add strength to our findings.

## Conclusions

Grandmothers across the four study settings expressed interest in oral PrEP and the ring for HIV prevention and a willingness to support their use by their pregnant or breastfeeding daughters and daughters-in law to protect their children and grandchildren. Both pregnant and breastfeeding women and male partners, in addition to grandmothers themselves, saw grandmothers as having a supportive and sometimes key influencer role in the health-related decision making of pregnant and breastfeeding women due to their own prior parity experience and their knowledge of traditional and cultural practices that could also be leveraged. Although the intensity of these roles differed by setting, these data are indicative that grandmothers' supportive influence may be extended to support uptake of and adherence to biomedical HIV prevention options and potentially contribute to the decline in HIV acquisition among pregnant and breastfeeding women in these communities.

## Supporting information

**S1 File. Interview guide grandmothers.**
(PDF)

**S2 File. Interview guide pregnant and breastfeeding women.**
(PDF)

**S3 File. Interview guide male partners.**
(PDF)

**S1 Checklist. COREQ checklist.**
(PDF)

## Acknowledgments

The MAMMA trial was designed and implemented by the Microbicide Trials Network (MTN). The authors are grateful to the study participants for their participation and dedication and thank the research site study team members, the MTN-041/MAMMA Protocol Management team, the MTN Leadership and Operations Center, Women's Global Health Imperative (WGHI) RTI International and FHI 360 for their contributions to data collection. The content is solely the responsibility of the authors. The rings and oral PrEP used as sample products were developed and supplied by the International Partnership for Microbicides (IPM) and Gilead Sciences respectively.

## Author Contributions

**Conceptualization:** Krishnaveni Reddy, Julia Ryan, Ariane van der Straten.

**Data curation:** Krishnaveni Reddy, Julia Ryan.

**Formal analysis:** Krishnaveni Reddy, Doreen Kemigisha, Miria Chitukuta, Sufia Dadabhai, Julia Ryan, Ariane van der Straten.

**Investigation:** Florence Mathebula, Siyanda Tenza.

**Writing – original draft:** Krishnaveni Reddy.

**Writing – review & editing:** Krishnaveni Reddy, Doreen Kemigisha, Miria Chitukuta, Sufia Dadabhai, Florence Mathebula, Siyanda Tenza, Thesla Palanee-Phillips, Julia Ryan, Nicole Macagna, Petina Musara, Ariane van der Straten.

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
