## [Decision Letter · Decision Letter 0]

29 Mar 2022

PONE-D-21-27994Ask the way from those who have walked it before – Grandmothers’ roles in health-related decision making and HIV pre-exposure prophylaxis use among pregnant and breastfeeding women in AfricaPLOS ONE

Dear Dr. Reddy,

Thank you for submitting your manuscript to PLOS ONE. After careful consideration, we feel that it has merit but does not fully meet PLOS ONE’s publication criteria as it currently stands. Therefore, we invite you to submit a revised version of the manuscript that addresses the points raised during the review process. Please address all of the reviewers' comments, with which I agree.

We look forward to receiving your revised manuscript.

Kind regards,

Douglas S. Krakower, MD

Academic Editor

PLOS ONE

Journal Requirements:

2. Please include additional information regarding the survey or interview guide used in the study and ensure that you have provided sufficient details that others could replicate the analyses. For instance, if you developed a questionnaire or interview guide as part of this study and it is not under a copyright more restrictive than CC-BY, please include a copy, in both the original language and English, as Supporting Information.

The MTN is funded by the National Institute of Allergy and Infectious Diseases (UM1AI068633, UM1AI068615, and UM1AI106707), with cofunding from the Eunice Kennedy Shriver National Institute of Child Health and Human Development and the National Institute of Mental Health, all components of the US National Institutes of Health. 

The MTN-041/MAMMA study was funded by the Division of AIDS, US National Institute of Allergy and Infectious Diseases (https://www.niaid.nih.gov/about/daids), US Eunice Kennedy Shriver National Institute of Child Health and Human Development (https://www.nichd.nih.gov/), US National Institute of Mental Health (https://www.nimh.nih.gov/), US National Institutes of Health (https://www.nih.gov/) (Grant numbers: UM1AI068633, UM1AI068615, UM1AI106707).  The funders played no role in the study design, data collection and analysis, decision to publish, or preparation of the manuscript.

Reviewers' comments:

Reviewer's Responses to Questions

**Comments to the Author**

1. Is the manuscript technically sound, and do the data support the conclusions?

Reviewer #1: Partly

Reviewer #2: Yes

2. Has the statistical analysis been performed appropriately and rigorously? 

Reviewer #1: N/A

Reviewer #2: Yes

3. Have the authors made all data underlying the findings in their manuscript fully available?

Reviewer #1: Yes

Reviewer #2: Yes

4. Is the manuscript presented in an intelligible fashion and written in standard English?

Reviewer #1: Yes

Reviewer #2: Yes

5. Review Comments to the Author

Reviewer #1: Thank you for the opportunity to review this interesting paper. Much of the paper is well written and most of your findings are clear. However, the paper needs to be motivated better and the discussion needs to move away from restating the results to telling us more about what is new and exciting about your work and what we can do with the information you present. I recommend looking at qualitative reporting guidelines (such as the COREQ) to ensure your procedures and analysis are detailed and correct. Below are more specific comments.

Introduction

Make sure you have a reference for each statement that you make e.g. line 73.

Please tell us why Grandmothers have not been studied as influencers. Line 77

In many cases you are using a / when you can write out the word. For example, line 79: Grandmothers/elders – A Grandmother is not necessarily the same as an elder. What if you are a Grandma at 40?

This happens a lot throughout the manuscript and is not the correct use of the /. Another example is line 115 - recorded and transcribed/translated. Transcription and translation are two very different things. It feels like you are cutting corners in writing full sentences which Im sure was not your intention.

Procedures

This section is not clear. You need to state the method used (FGDs) from the start and then move on to the details. At the moment is reads like you used questionnaires to collect the FGD data, but that isn’t the case. There is no information on who collected the data, how the tools were developed and what training took place. Please take a look at some qualitative reporting guidelines - such as the COREQ – and review and rewrite this section.

Analysis

Why did you follow the SEM? It doesn’t feature anywhere else in the manuscript. And where do codes such as Pill, Ring and Preference fit in the SEM?

How did you analyse the assessments?

What approach did you use (inductive/deductive/hybrid)?

Results

The description of the demographic data provided in the tables is long and not necessary. That’s what the tables are for (and the tables are very good and clear).

A summary paragraph of the results – at the start of the results – would be useful

158 - Pregnant and breastfeeding women were assessed about their views on who, besides – Please make it clearer when you are describing data from the assessments and then the FGDs.

There is a lot in the results. I think you need to chose one approach of presenting the findings and stick to it.

Discussion

Most of the discussion reads like a restatement of the results. There is not enough to show where this work sits in existing literature, and how it adds to the gap. Apart from mentioning that Grandmothers can be harnessed to help with PrEP and ring use, we aren’t really given any concrete examples of what this means. Tell the reader why what you have found is important and how it can be used.

Reviewer #2: OVERALL

This is a fascinating study on an novel field in need of culturally-aware research. With the edits and clarifications I have listed throughout this review, I believe this paper will provide insight that can help to normalize PrEP use in young women across Sub-Saharan Africa.

GRAMMAR

Please copy-edit grammar and punctuation throughout the paper, as there are some errors and inconsistencies (I have not listed every single one in this review). I recommend avoid passive voice throughout the paper. Finally, try to minimize casual punctuation, like parentheticals and slashes, when it can be better communicated with conjunctions.

TITLE

Since “Ask the way from those who have walked it before” is a quote from one of the respondents, I suggest putting it in quotation marks.

Can you specify that this takes place in Sub-Saharan Africa instead of just Africa?

FUNDING STATEMENT

Please include initials of the authors who received each grant.

ABSTRACT

Line 35-36 - Please clarify that you are talking about mothers of the pregnant/breastfeeding women, not grandmothers of the pregnant/breastfeeding women.

Line 42 - Please specify what you mean by “other groups”?

Can you clarify more of the methods, particularly that you used focus groups, and what the sample sizes were? Would you consider this a purely qualitative study or a mixed-methods study?

Line 44 - Did you specifically ask grandmothers’ about how they could support PrEP uptake or adherence? It is important to distinguish between the two, as they are separate issues.

INTRODUCTION

Lines 53-54 - I suggest including literature about why the HIV incidence during pregnancy is so high (both behavioral and biological characteristics). Here are some relevant citations:

Moodley D, Moodley P, Sebitloane M, Soowamber D, McNaughton-Reyes HL, Groves AK, et al. High Prevalence and Incidence of Asymptomatic Sexually Transmitted Infections During Pregnancy and Postdelivery in KwaZulu Natal, South Africa. Sex Transm Dis. 2015;42(1): 43–47. pmid:25504300

Kinuthia J, Drake AL, Matemo D, Richardson BA, Zeh C, Osborn L, et al. HIV acquisition during pregnancy and postpartum is associated with genital infections and partnership characteristics. AIDS. 2015;29(15): 2025–2033. pmid:26352880

Line 68 - Please clarify whether a woman can replace the ring herself and/or hold multiple months’ supply at home, or whether she needs to return to a clinic every month. There are challenges to continuous prevention methods like this, as well as oral medications.

Line 68 - I am a bit concerned that the PrEP ring safety trials during pregnancy are “ongoing”. Please clarify whether it is currently medically recommended to use the ring during pregnancy.

Line 79 - Please define the term ‘elders’ and how they relate to grandmothers.

As PrEP stigma comes up in the data, may I suggest adding a sentence or two about this and other barriers to PrEP uptake and adherence in young women (line 70)? This will better frame WHY grandmothers are needed as a support system.

METHODS

All tables are cut off in the PDF view. I suggest making them narrower, or rotating them 90 degrees.

Line 106 - The language is called ‘isiZulu’, whereas typically ‘Zulu’ refers to the culture.

Line 121 - Please cite the socio-ecological framework and describe how you applied it to the codebook. Did you consider it when developing the results? If so, elaborate later in the paper, as it is never mentioned again.

Lines 140-152 - Would you say the unique results coming from the South African population was representative of the population as a whole, or due to your recruitment methods/locations?

Table 2 - Please include a footnote when the % does not add up to 100%, stating that respondents could select multiple answers.

Analysis section - What software was used to analyze? How many coders were there, and did they participate in data collection? Did you ensure inter-coder reliability? Speak more to the aspects of qualitative analysis laid out in the CORE-Q guidelines.

RESULTS

The results are well structured, with a good balance of tables, free quotes, and embedded quotes. It is quite long and repetitive at times, but tells a clear and convincing story.

You have collected a large quantity of high quality data from a variety of key settings. One issue with such variance (in terms of nationality, gender, age, and which form of PrEP is being discussed) is that the context for certain quotes is sometimes unclear. Please ensure you are specific when talking about any cultural practices about which countries’ respondents mentioned them, so as not to conflate different African cultures. Further, always specify which form of PrEP is being discussed when it is unclear.

Again, all tables are cut off in the PDF view. I suggest making them narrower, or turning them 90 degrees.

The wording around Table 3, starting in Line 159, is unclear. Please clarify that you asked women who has a greater impact on their health-related decisions, and clarify what is being influenced in the the title of Table 3.

Line 169 - Does “elders” exclusively refer to grandmothers, or to older individuals in the community?

Line 177 - Please make it clear whether the Lobola was specifically talked about in the South African context, or whether any other countries’ respondents discussed it.

Table 4 - Under “emotional supportive role”, you should change the identifier “breastfeeding women” to “breastfeeding woman”(singular rather than plural).

Line 339- Change “western medicine” to ‘biomedical pharmaceuticals’ or something similar. If you choose to keep it, ‘Western’ should be capitalized.

Lines 354, 355, 357 - Hyphenate ‘HIV-negative’ and ‘HIV-positive’

DISCUSSION

Line 373 - Avoid passive voice here and throughout the paper

Lines 390-393 - Very long sentence, split into two.

Lines 395 - 398 - Very long sentence, split into two.

Line 399, Line 410- Space before parentheticals are missing.

Would you recommend one form of PrEP over the other? Would you recommendations differ between settings?

What are next steps to engage grandmothers in PrEP rollout? Are there any interventions that have engaged grandmothers in other aspects of women’s health in these or similar populations?

LIMITATIONS

Can you say how you mitigated the third and fourth limitations, if at all?

CONCLUSION

Line 438 - Be careful when extrapolating these findings of potentially increased PrEP uptake, to lower rates of HIV acquisition (which also requires adherence to PrEP). Make it clear throughout the paper whether you are speaking about PrEP adherence or uptake.

Line 436 - “With the right framing and approach” - I am curious to hear your suggestions about this approach (maybe in the Discussion, as I mention above).

6. PLOS authors have the option to publish the peer review history of their article (what does this mean?). If published, this will include your full peer review and any attached files.

Reviewer #1: No

Reviewer #2: **Yes: **Scarlett Bergam

---

## [Author Response · Author response to Decision Letter 0]

5 Jun 2022

Dear reviewers

Re: List of responses and changes

Thank you for your review and feedback regarding our manuscript listed below. Please see responses to reviewers' comments in the table below as well as tracked changed document submitted. Please note that location of changes indicated below are with reference to the tracked changed version of the manuscript.

Manuscript Title: ‘‘Ask the way from those who have walked it before” – Grandmothers’ roles in health-related decision making and HIV pre-exposure prophylaxis use among pregnant and breastfeeding women in sub-Saharan Africa

Reviewers' comments:

Author’s response: Manuscript has been reviewed and updated in alignment with PLOS ONES’s style requirements

2. Please include additional information regarding the survey or interview guide used in the study and ensure that you have provided sufficient details that others could replicate the analyses. For instance, if you developed a questionnaire or interview guide as part of this study and it is not under a copyright more restrictive than CC-BY, please include a copy, in both the original language and English, as Supporting Information.

Author’s response: 

Procedures (Lines 151 – 155) has been updated to include the following text (see italicized text): Gender-matched trained local social scientists fluent in local languages then facilitated the FGDs using semi-structured guides. A second trained staff member was present to assist with note taking. The FGD guides consisted of an introduction where the facilitators explained the goals and rules of the FGD followed by open-ended questions and prompts to guide the discussion. These guides were developed and pilot tested by the research team for each stakeholder group

Further detail regarding the guide is included in Lines 156 – 159: Topics discussed included HIV risk perceptions, cultural beliefs and practices relating to pregnancy and breastfeeding, health-related decision making, key influencers and interest in two new HIV prevention products while pregnant or breastfeeding: daily oral PrEP pills and the monthly vaginal ring. 

Copies of the three guides (English) used to facilitate FGDs with the different stakeholder groups (grandmothers, pregnant and breastfeeding women and male partners) have been included as supporting documentation (S1, S2 and S3)

Author’s response: The correct information is as follows and has been updated to match in both the “Funding Information” and “Financial Disclosure” sections: The MAMMA study was designed and implemented by the Microbicide Trials Network (MTN). The MTN is funded by the National Institute of Allergy and Infectious Diseases (UM1AI068633, UM1AI068615, UM1AI106707), with co-funding from the Eunice Kennedy Shriver National Institute of Child Health and Human Development and the National Institute of Mental Health, all components of the U.S. National Institutes of Health. The content is solely the responsibility of the authors and does not necessarily represent the official views of the National Institutes of Health.

The MTN is funded by the National Institute of Allergy and Infectious Diseases (UM1AI068633, UM1AI068615, and UM1AI106707), with cofunding from the Eunice Kennedy Shriver National Institute of Child Health and Human Development and the National Institute of Mental Health, all components of the US National Institutes of Health. 

The MTN-041/MAMMA study was funded by the Division of AIDS, US National Institute of Allergy and Infectious Diseases (https://www.niaid.nih.gov/about/daids), US Eunice Kennedy Shriver National Institute of Child Health and Human Development (https://www.nichd.nih.gov/), US National Institute of Mental Health (https://www.nimh.nih.gov/), US National Institutes of Health (https://www.nih.gov/) (Grant numbers: UM1AI068633, UM1AI068615, UM1AI106707). The funders played no role in the study design, data collection and analysis, decision to publish, or preparation of the manuscript.

Author’s response: Funding-related text have been removed from the manuscript

Acknowledgments (Lines 587-597) has been updated as follows: The MAMMA trial was designed and implemented by the Microbicide Trials Network (MTN). The authors are grateful to the study participants for their participation and dedication and thank the research site study team members, the MTN-041/MAMMA Protocol Management team, the MTN Leadership and Operations Center, Women’s Global Health Imperative (WGHI) RTI International and FHI 360 for their contributions to data collection. The content is solely the responsibility of the authors. The rings and oral PrEP used as sample products were developed and supplied by the International Partnership for Microbicides (IPM) and Gilead Sciences respectively.

The correct funding statement/financial disclosure is as follows: The MAMMA study was designed and implemented by the Microbicide Trials Network (MTN). The MTN is funded by the National Institute of Allergy and Infectious Diseases (UM1AI068633, UM1AI068615, UM1AI106707), with co-funding from the Eunice Kennedy Shriver National Institute of Child Health and Human Development and the National Institute of Mental Health, all components of the U.S. National Institutes of Health. The content is solely the responsibility of the authors and does not necessarily represent the official views of the National Institutes of Health.

Reviewer #1: 

Reviewers' comments:

1. Thank you for the opportunity to review this interesting paper. Much of the paper is well written and most of your findings are clear. However, the paper needs to be motivated better and the discussion needs to move away from restating the results to telling us more about what is new and exciting about your work and what we can do with the information you present. I recommend looking at qualitative reporting guidelines (such as the COREQ) to ensure your procedures and analysis are detailed and correct. Below are more specific comments.

Author’s response: Thank you for your comment and recommendations to improve our manuscript We have addressed the specific comments as raised below and included the completed COREQ checklist as a supplementary document (S4)

INTRODUCTION:

2. Make sure you have a reference for each statement that you make e.g. line 73.

Author’s response: 

Introduction (Lines 107-108) has been updated as follows: Grandmothers are family members who play a central role in the sub-Saharan African family, as sources of information, wisdom and comfort (28).

Michel J, Stuckelberger A, Tediosi F, Evans D, van Eeuwijk P. The roles of a Grandmother in African societies - please do not send them to old people's homes. Journal of global health. 2020;10(1):010361.

3. Please tell us why Grandmothers have not been studied as influencers. Line 77

Author’s response: 

Introduction (Lines 102 - 106) has been updated as follows: Strategies to improve PrEP use to date have mainly been focused on the user or on engaging male partner(s) and the role of family support in this regard has not been extensively researched. This is likely due to the sensitive nature of HIV prevention and fear of stigma and judgement. Family support has however been determined to have a positive impact on patients’ abilities to self-manage chronic conditions by influencing their daily behavior (27) and has potential to be extended to HIV prevention particularly among pregnant and breastfeeding women.

4. In many cases you are using a / when you can write out the word. For example, line 79: Grandmothers/elders – A Grandmother is not necessarily the same as an elder. What if you are a Grandma at 40?

This happens a lot throughout the manuscript and is not the correct use of the /. Another example is line 115 - recorded and transcribed/translated. Transcription and translation are two very different things. It feels like you are cutting corners in writing full sentences which Im sure was not your intention.

Author’s response: 

Introduction (Line 114): The text “/elders” has been deleted 

Procedures (Lines 162-163): Text has been updated as follows: FGDs lasted ~ two and a half hours (Minimum one hour and maximum three and a half hours) and were audio recorded, translated and transcribed in English as applicable. 

The manuscript was reviewed and updated to correct the use of “/”. 

PROCEDURES: 

3. This section is not clear. You need to state the method used (FGDs) from the start and then move on to the details. At the moment is reads like you used questionnaires to collect the FGD data, but that isn’t the case. There is no information on who collected the data, how the tools were developed and what training took place. Please take a look at some qualitative reporting guidelines - such as the COREQ – and review and rewrite this section.

Author’s response: Thank you for this guidance. Please see provided completed COREQ checklist (S4). I have edited the text as per below: 

Procedures (Lines 146 - 164): Data was collected at the four research clinic study sites. All participants provided written informed consent before demographic information was individually collected by site staff through the use of structured questionnaires in local languages (Chichewa in Blantyre, Malawi; isiZulu or English in Johannesburg, South Africa; Luganda in Kampala, Uganda and Shona in Chitungwiza, Zimbabwe). A staff administered behavioural assessment via structured questionnaire was also completed with the pregnant and breastfeeding women and male participants. Gender-matched trained local social scientists fluent in local languages then facilitated the FGDs using semi-structured guides. A second trained staff member was present to assist with note taking. The FGD guides consisted of an introduction where the facilitators explained the goals and rules of the FGD followed by open-ended questions and prompts to guide the discussion. These guides were developed and pilot tested by the research team for each stakeholder group. Topics discussed included HIV risk perceptions, cultural beliefs and practices relating to pregnancy and breastfeeding, health-related decision making, key influencers and interest in two new HIV prevention products while pregnant or breastfeeding: daily oral PrEP pills and the monthly vaginal ring (11, 34). Participants viewed a four-minute educational video (in the local language) and handled sample products immediately prior to discussing these new HIV prevention options. Participants were requested to use pseudonyms during the FGDs to protect their identities. FGDs lasted ~ two and a half hours (Minimum one hour and maximum three and a half hours) and were audio recorded, translated and transcribed in English as applicable. Facilitators completed a debriefing summary report after each FGD for rapid thematic analysis.

ANALYSIS:

4. Why did you follow the SEM? It doesn’t feature anywhere else in the manuscript. And where do codes such as Pill, Ring and Preference fit in the SEM? How did you analyse the assessments? What approach did you use (inductive/deductive/hybrid)?

Author’s response: 

Analysis (Lines 167-190) has been updated as follows: Demographic and behavioural data are presented descriptively by country. Fisher’s exact tests were used to calculate differences by country with regard to women’s responses on who has the most influence on their decisions during pregnancy and while breastfeeding besides themselves. For the qualitative data, analysis workshops were held for all site staff involved in qualitative data collection to conduct a preliminary analysis of the data; workshops directly informed the iterative development of the codebook used to systematically analyse all qualitative data. The codebook for this study followed a socio-ecological framework that was adapted to include the spheres of influences on future use of HIV PrEP during pregnancy and breastfeeding. This included the mother and baby dyad and the male partner or father of the baby, followed by family members (mostly grandmother of the baby, siblings and other family members), institutional and socio- structural factors (33). FGD transcripts were coded by four data analysts using Dedoose software (v7.0.23). An acceptable level of intercoder reliability was set and maintained at approximately 80% agreement. The analysis team met weekly to discuss coding questions, issues and emerging themes and resolve discrepancies. Coded data reports were further summarized thematically into analytical memos that were reviewed by site teams (33, 35). For this analysis, we looked at family influences specifically grandmothers as they emerged as being important influencers from responses to structured questionnaires and FGDs. Data coded for “FAMILY” were extracted from all FGD transcripts and stratified by stakeholder group type (i.e., grandmothers, pregnant or breastfeeding women and male partners) in addition to country. Additionally, a product-focused acceptability framework (36) was used to understand prospective acceptability of the two HIV prevention methods. For this, data coded for “PILL”, “RING” and “PREFERENCE” were extracted from Grandmother FGD transcripts and stratified by country. Data reports were then thematically analysed by representatives of the four research clinic study sites into analytical memos that were reviewed by the writing team biweekly to discuss coding questions and emerging themes. 

We used a hybrid mixed method approach for this analysis

Introduction (Line 112-114) has been updated as follows: We address this gap by drawing on mixed method data gathered during the multi-site MTN-041/MAMMA study in SSA (33).

RESULTS:

5. The description of the demographic data provided in the tables is long and not necessary. That’s what the tables are for (and the tables are very good and clear).

Author’s response: Text has been edited to be more concise as per below:

Results (Line 201-214): Demographic data are presented descriptively by stakeholder group and country in Table 1. The mean age of grandmothers was 50 years (min 36, max 69) with most grandmothers living with their children (81%, N=55). The mean age of pregnant and breastfeeding women and male partners was 27 years (min 19, max 40) and 31 years (min 19, max 54) respectively. The South African pregnant and breastfeeding women and male partners differed from those in the other settings with regards to marital status and living arrangements. Most were single (93% of pregnant or breastfeeding women and 92% of male partners) and majority (67% of pregnant or breastfeeding women [N=10] and 58% of male partners [N=7]) were living with adult family members including parents and siblings. In the other settings, most pregnant and breastfeeding women and male partners were married (83%–94%) and living with their spouse or primary partner (79%–94%). 

6. A summary paragraph of the results – at the start of the results – would be useful

Author’s response: A summary paragraph of the results has been added to the start of the results section following the demographic data as per below:

Results (Lines 234 – 240): Overall, data collected from the FGDs described grandmothers as important sources of information, playing both supportive and influencer roles, due to personal maternal experience and generational knowledge. All stakeholder groups agreed that HIV prevention related decision making should be made by pregnant and breastfeeding women themselves or together with partners. However there was indication from the pregnant and breastfeeding women group that grandmothers could be involved in HIV prevention product use in a supportive role if this was disclosed to them. Importantly, grandmothers themselves expressed willingness to support PrEP use.

7. 158 - Pregnant and breastfeeding women were assessed about their views on who, besides – Please make it clearer when you are describing data from the assessments and then the FGDs.

Author’s response: 

Results (Lines 243-248) has been updated as follows: Data from the behavioural assessment (Table 2) indicated that the majority of pregnant and breastfeeding women in Malawi, Uganda and Zimbabwe, thought that besides themselves, the father of the baby had the most influence on their decisions during pregnancy (60%–88%) and while breastfeeding (53%-92%). 

8. There is a lot in the results. I think you need to chose one approach of presenting the findings and stick to it.

Author’s response: Please clarify what is meant by choosing one approach of presenting the findings. The second reviewer has indicated the results are well structured, with a good balance of tables, free quotes, and embedded quotes but is long and repetitive at times. We have edited the results section to reduce the repetitiveness and amount of text.

DISCUSSION:

9. Most of the discussion reads like a restatement of the results. There is not enough to show where this work sits in existing literature, and how it adds to the gap. Apart from mentioning that Grandmothers can be harnessed to help with PrEP and ring use, we aren’t really given any concrete examples of what this means. Tell the reader why what you have found is important and how it can be used.

Author’s response: The discussion has been reviewed and updated in accordance with the reviewers recommendations. The following text has also been added: 

Discussion (Lines 539-550): These data bring to light new insights and opportunities to potentially utilize grandmothers to support PrEP uptake and use particularly among pregnant and breastfeeding women. Efforts have been made to enhance the skills of grandmothers to discuss issues related to sex and sexuality with young girls to reduce unintended pregnancy (41) as well as HIV acquisition (42) and their support has been shown to be important for prevention. It is therefore possible that their supportive roles during pregnancy and breastfeeding can be similarly harnessed to promote HIV prevention product uptake and adherence among their daughters and daughters-in-law. Further consultations with grandmothers on how to best engage them, with a focus on PrEP education and adherence support mechanisms and strategies, are needed.

Reviewer #2:

Reviewers' comments:

This is a fascinating study on an novel field in need of culturally-aware research. With the edits and clarifications I have listed throughout this review, I believe this paper will provide insight that can help to normalize PrEP use in young women across Sub-Saharan Africa.

Author’s response: Thank you for your comment and recommendations to improve our manuscript

GRAMMAR:

1. Please copy-edit grammar and punctuation throughout the paper, as there are some errors and inconsistencies (I have not listed every single one in this review). I recommend avoid passive voice throughout the paper. Finally, try to minimize casual punctuation, like parentheticals and slashes, when it can be better communicated with conjunctions.

Author’s response: We have reviewed the manuscript and edited to address the concerns regarding use of passive voice, grammar and punctuation.

TITLE:

1. Since “Ask the way from those who have walked it before” is a quote from one of the respondents, I suggest putting it in quotation marks.

Author’s response: The title has been updated as per reviewer’s suggestion

Title page (Lines 2-4): ‘‘Ask the way from those who have walked it before” – Grandmothers’ roles in health-related decision making and HIV pre-exposure prophylaxis use among pregnant and breastfeeding women in sub-Saharan Africa

2. Can you specify that this takes place in Sub-Saharan Africa instead of just Africa?

Author’s response: The title has been updated as per reviewer’s suggestion

Title page (Lines 2-4): ‘‘Ask the way from those who have walked it before” – Grandmothers’ roles in health-related decision making and HIV pre-exposure prophylaxis use among pregnant and breastfeeding women in sub-Saharan Africa

FUNDING STATEMENT:

3. Please include initials of the authors who received each grant.

Author’s response: The grant was received by the Microbicide Trials Network and not by a specific author

ABSTRACT:

4. Line 35-36 - Please clarify that you are talking about mothers of the pregnant/breastfeeding women, not grandmothers of the pregnant/breastfeeding women.

Author’s response: Text has been updated as follows 

Abstract (Lines 39-43): During the MTN-041/MAMMA study, we explored the influence of grandmothers (mothers and mothers-in-law of pregnant and breastfeeding women) in eastern and southern Africa on the health-related decisions of pregnant and breastfeeding women and their potential to support use of HIV prevention products.

5. Line 42 - Please specify what you mean by “other groups”?

Author’s response: This has been clarified as follows 

Abstract (Lines 43-46): To do this we used structured questionnaires and focus group discussions with three stakeholder groups: 1) grandmothers, 2) HIV-uninfected currently or recently pregnant or breastfeeding women and 3) male partners of currently or recently pregnant or breastfeeding women.

Abstract (Lines 52-54): Grandmothers expressed willingness to support pre-exposure prophylaxis use and agreed with the other two stakeholder groups that this decision should be made by women themselves or together with partners.

6. Can you clarify more of the methods, particularly that you used focus groups, and what the sample sizes were? Would you consider this a purely qualitative study or a mixed-methods study?

Author’s response: Text has been updated as follows:

Abstract (Lines 43-48): To do this we used structured questionnaires and focus group discussions with three stakeholder groups : 1) grandmothers, 2) HIV-uninfected currently or recently pregnant or breastfeeding women and 3) male partners of currently or recently pregnant or breastfeeding women. A total of 23 focus group discussions comprising 68 grandmothers, 65 pregnant or breastfeeding women and 63 male partners were completed across four study sites. 

Study Design (Lines 120-123): MAMMA (Microbicide/PrEP Acceptability among Mothers and Male Partners in Africa) was an exploratory, mixed method study conducted between May and November 2018 at four research clinic study sites in the following settings: Blantyre (Malawi); Johannesburg (South Africa); Kampala (Uganda) and Chitungwiza (Zimbabwe). 

7. Line 44 - Did you specifically ask grandmothers’ about how they could support PrEP uptake or adherence? It is important to distinguish between the two, as they are separate issues.

Author’s response: The framing of the questions in the FGD guide were geared around supporting the general use of PrEP. PrEP uptake or adherence was not specifically stated however data from all group discussions indicated that the decision to use PrEP (PrEP uptake) should be made by pregnant or breastfeeding women alone or together with their partners. Some pregnant and breastfeeding women mentioned that grandmothers could support PrEP use once they are already taking it (PrEP adherence) as did grandmothers. I have updated the text as follows: 

Abstract (Lines 50-57): While pregnant and breastfeeding women were not keen to involve grandmothers in HIV prevention decision making, they were accepting of grandmothers’ involvement in a supportive role. Grandmothers expressed willingness to support pre-exposure prophylaxis use and agreed with the other two stakeholder groups that this decision should be made by women themselves or together with partners. These novel data indicate potential for grandmothers’ health related supportive roles to be extended to support decision-making and adherence to biomedical HIV prevention options, and possibly contribute to the decline in HIV acquisition among pregnant and breastfeeding women in these communities.

INTRODUCTION:

8. Lines 53-54 - I suggest including literature about why the HIV incidence during pregnancy is so high (both behavioral and biological characteristics). Here are some relevant citations:

Moodley D, Moodley P, Sebitloane M, Soowamber D, McNaughton-Reyes HL, Groves AK, et al. High Prevalence and Incidence of Asymptomatic Sexually Transmitted Infections During Pregnancy and Postdelivery in KwaZulu Natal, South Africa. Sex Transm Dis. 2015;42(1): 43–47. pmid:25504300

Kinuthia J, Drake AL, Matemo D, Richardson BA, Zeh C, Osborn L, et al. HIV acquisition during pregnancy and postpartum is associated with genital infections and partnership characteristics. AIDS. 2015;29(15): 2025–2033. pmid:26352880

Author’s response: Text related to why HIV incidence is high during pregnancy as well as the provided references have been added as per below

Introduction (Lines 65-68): Potential mechanisms for increased HIV susceptibility during these maternal periods include both biological and behavioral factors such as hormonal changes that affect the genital tract mucosal surfaces or immune responses (3), high rates of asymptomatic sexually transmitted infections (2) and partner HIV infection through multiple concomitant partnerships (4).

9. Line 68 - Please clarify whether a woman can replace the ring herself and/or hold multiple months’ supply at home, or whether she needs to return to a clinic every month. There are challenges to continuous prevention methods like this, as well as oral medications.

Author’s response: Clarification has been added as follows:

Introduction (Lines 88-90): The open label extensions additionally demonstrated that women can store multiple months’ ring supply in their homes and replace the ring themselves (15, 16).

10. Line 68 - I am a bit concerned that the PrEP ring safety trials during pregnancy are “ongoing”. Please clarify whether it is currently medically recommended to use the ring during pregnancy.

Author’s response: Clarification has been added as follows:

Introduction (Lines 90-93): The ring is not currently recommended for pregnant and breastfeeding women’s use as studies to determine the safety and acceptability of the ring during pregnancy and breastfeeding are still ongoing (17); however ring use in the periconception period was not associated with adverse effects on pregnancy or infant outcomes (18). 

11. Line 79 - Please define the term ‘elders’ and how they relate to grandmothers.

Author’s response: 

Introduction (Line 114): The text “/elders” has been deleted 

12. As PrEP stigma comes up in the data, may I suggest adding a sentence or two about this and other barriers to PrEP uptake and adherence in young women (line 70)? This will better frame WHY grandmothers are needed as a support system.

Author’s response: Text has been added as follows:

Introduction (Lines 96-99): Barriers to PrEP use include inconsistent access to PrEP services, non-disclosure to male partners, provider bias, stigma related to HIV and PrEP use, PrEP cost, individual risk perception, low PrEP awareness, lack of social support for PrEP use, side effects as well as contextual factors such as gender and culture (24-26). 

METHODS

13. All tables are cut off in the PDF view. I suggest making them narrower, or rotating them 90 degrees. 

Author’s response: We were unable to narrow Table 1 without compromising its format. Also the PLOS ONE Table guidelines indicate: Tables do not have strict width and height requirements. Do not split your table or otherwise try to make the table appear within the manuscript margins if it does not fit on one page. In Word, tables that run off of the manuscript page can be seen using Draft View. In the PDF version of the published article, very wide tables may be printed sideways, and long tables may span more than one page.

We have narrowed Tables 2, 3 and 4

14. Line 106 - The language is called ‘isiZulu’, whereas typically ‘Zulu’ refers to the culture.

Author’s response: Text has been added as follows:

Procedures (Lines 149): Text has been updated to isiZulu

15. Line 121 - Please cite the socio-ecological framework and describe how you applied it to the codebook. Did you consider it when developing the results? If so, elaborate later in the paper, as it is never mentioned again.

Author’s response: Text has been updated as follows:

Analysis (Lines 172-186): The codebook for this study followed a socio-ecological framework that was adapted to include the spheres of influences on future use of HIV PrEP during pregnancy and breastfeeding. This included the mother and baby dyad and the male partner or father of the baby, followed by family members (mostly grandmother of the baby, siblings and other family members), institutional and socio- structural factors (33). FGD transcripts were coded by four data analysts using Dedoose software (v7.0.23). An acceptable level of intercoder reliability was set and maintained at approximately 80% agreement. The analysis team met weekly to discuss coding questions, issues and emerging themes and resolve discrepancies. Coded data reports were further summarized thematically into analytical memos that were reviewed by site teams (33, 35). For this analysis, we looked at family influences specifically grandmothers as they emerged as being important influencers from responses to structured questionnaires and in FGDs. Data coded for “FAMILY” were extracted from all FGD transcripts and stratified by stakeholder group type (i.e., grandmothers, pregnant or breastfeeding women and male partners) in addition to country. Additionally, a product-focused acceptability framework (36) was used to understand prospective acceptability of the two HIV prevention methods. For this, data coded for “PILL”, “RING” and “PREFERENCE” were extracted from Grandmother FGD transcripts and stratified by country. 

16. Lines 140-152 - Would you say the unique results coming from the South African population was representative of the population as a whole, or due to your recruitment methods/locations?

Author’s response: This would be difficult to speculate as the study participants are representative of the population in Hillbrow, Johannesburg as stated under Study population and settings and Hillbrow serves as a port of entry for migrants and immigrants from the townships and rural areas of other South African provinces as well as the rest of Africa. We did however use a diversity of recruitment methods and participants were recruited from a variety of locations 

17. Table 2 - Please include a footnote when the % does not add up to 100%, stating that respondents could select multiple answers.

Author’s response: The above table is now table 1. A footnote indicating the above has been added to the table

18. Analysis section - What software was used to analyze? How many coders were there, and did they participate in data collection? Did you ensure inter-coder reliability? Speak more to the aspects of qualitative analysis laid out in the CORE-Q guidelines

Author’s response: This information has been previously published and was cited within the analysis section. This additional information has now been added per below. I have also included the completed COREQ checklist as supporting information (S4).

Analysis (Lines 176-180): FGD transcripts were coded by four data analysts using Dedoose software (v7.0.23). An acceptable level of intercoder reliability was set and maintained at approximately 80% agreement. The analysis team met weekly to discuss coding questions, issues and emerging themes and resolve discrepancies. Coded data reports were further summarized thematically into analytical memos that were reviewed by site teams (33, 35). 

RESULTS:

19. The results are well structured, with a good balance of tables, free quotes, and embedded quotes. It is quite long and repetitive at times but tells a clear and convincing story.

Author’s response: Thank you. We have reviewed and edited the Results section to be more concise.

20. You have collected a large quantity of high quality data from a variety of key settings. One issue with such variance (in terms of nationality, gender, age, and which form of PrEP is being discussed) is that the context for certain quotes is sometimes unclear. Please ensure you are specific when talking about any cultural practices about which countries’ respondents mentioned them, so as not to conflate different African cultures. Further, always specify which form of PrEP is being discussed when it is unclear.

Author’s response: We have reviewed the manuscript and addressed the above

21. Again, all tables are cut off in the PDF view. I suggest making them narrower, or turning them 90 degrees.

Author’s response: We were unable to narrow Table 1 without compromising its format. Also, the PLOS ONE Table guidelines indicate: Tables do not have strict width and height requirements. Do not split your table or otherwise try to make the table appear within the manuscript margins if it does not fit on one page. In Word, tables that run off of the manuscript page can be seen using Draft View. In the PDF version of the published article, very wide tables may be printed sideways, and long tables may span more than one page.

We have narrowed Tables 2, 3 and 4

22. The wording around Table 3, starting in Line 159, is unclear. Please clarify that you asked women who has a greater impact on their health-related decisions, and clarify what is being influenced in the the title of Table 3.

Author’s response: Table 3 is now Table 2. Text has been amended as follows:

Results (Lines 243-248): Data from the behavioural assessment (Table 2) indicated that the majority of pregnant and breastfeeding women in Malawi, Uganda and Zimbabwe, thought that besides themselves, the father of the baby had the most influence on their decisions during pregnancy (60%–88%) and while breastfeeding (53%-92%).

Table 2’s title has been updated as follows: Pregnant and breastfeeding women’s responses when asked about who has the most influence on their decisions during pregnancy and while breastfeeding besides themselves

23. Line 169 - Does “elders” exclusively refer to grandmothers, or to older individuals in the community?

Author’s response: The term “elders” has been replaced with “grandmothers” as per below:

Results (Lines 255-258): South African grandmothers, pregnant and breastfeeding women, and male partners emphasized that it is the grandmothers (mother of the pregnant or breastfeeding woman) who make decisions, especially in cases where the pregnant or breastfeeding women live with their mothers or returned home to their mothers to give birth (even if married).

24. Line 177 - Please make it clear whether the Lobola was specifically talked about in the South African context, or whether any other countries’ respondents discussed it.

Author’s response: Lobola was mentioned by both South African and Zimbabwean participants. This has been clarified in the text as follows: 

Results (Lines 264-267): Decision making may also be impacted by lobola (payment a male partner or head of his family gives to the woman’s family in gratitude for allowing the marriage) or damages (payment made if a woman is impregnated before marriage to show that the male partner’s family accepts the baby as their own) as expressed by male partners and pregnant or breastfeeding women in South Africa as well as male partners in Zimbabwe.

I have also updated the quote to a shorter quote from a Zimbabwean male partner (Lines 268-273): I am always afraid of complications when you have not yet paid lobola to your in laws... You won’t have any say in your relationship. [Tinashe, Male partner, 19, Zimbabwe]

25. Table 4 - Under “emotional supportive role”, you should change the identifier “breastfeeding women” to “breastfeeding woman”(singular rather than plural).

Author’s response: Table 4 is now Table 3. The identifier “breastfeeding women” under “emotional supportive role” has been amended to “breastfeeding woman”.

26. Line 339- Change “western medicine” to ‘biomedical pharmaceuticals’ or something similar. If you choose to keep it, ‘Western’ should be capitalized.

Author’s response: Text has been amended as follows:

Results (Lines 437- 439): They did not feel that the bitterness of oral PrEP was culturally problematic as other biomedical interventions used during pregnancy are bitter (e.g., Fansidar for malaria) and because traditional medicines can also be bitter. 

27. Lines 354, 355, 357 - Hyphenate ‘HIV-negative’ and ‘HIV-positive’

Author’s response: This has been amended as per request per below:

Results (Lines 439 – 444): One grandmother likened HIV-negative pregnant women taking oral PrEP to HIV-positive pregnant women taking ARVs, which is widely accepted: There is no cultural belief that can prevent a pregnant woman from taking Truvada because the Government has a policy that all pregnant women who are found to be HIV-positive at the ANC, when the pregnancy is term, they are given drugs to take in order to prevent the unborn baby from contracting HIV and AIDS. So I feel this drug called Truvada is like the same as that drug [for treatment]. [Tadala, Grandmother, 40, Malawi]

DISCUSSION:

28. Line 373 - Avoid passive voice here and throughout the paper

Author’s response: The manuscript has been reviewed and updated as requested. This particular text has been deleted

29. Lines 390-393 - Very long sentence, split into two.

Author’s response: This has been updated as follows 

Discussion (Lines 499-502): Grandmothers appeared to readily understand the purpose of the ring and oral PrEP with only a brief introduction (short video and sample product handling) and expressed a high level of willingness to support their pregnant and breastfeeding daughters and daughters-in-law in their use of these products. 

30. Lines 395 - 398 - Very long sentence, split into two.

Author’s response: This has been updated as follows 

Discussion (Lines 502-527): They expressed the need to protect their children and unborn grandchildren consequently preventing themselves from returning to a mothering role in their old age and serving as caregivers for their grandchildren orphaned by HIV, a frequent occurrence of the pandemic in SSA (37, 38). 

31. Line 399, Line 410- Space before parentheticals are missing.

Author’s response: This has been updated as requested in Discussion (Lines 527 and 542)

32. Would you recommend one form of PrEP over the other? Would you recommendations differ between settings?

Author’s response: The MAMMA study was initial research prior to conducting larger trials with pregnant and breastfeeding women (That are still in implementation) and was not meant to recommend one form of PrEP over another. Literature does however indicate the need for choices for HIV prevention, similar to contraceptives.

33. What are next steps to engage grandmothers in PrEP rollout? Are there any interventions that have engaged grandmothers in other aspects of women’s health in these or similar populations?

Author’s response: Text has been updated as follows 

Discussion (Lines 539-550): These data bring to light new insights and opportunities to potentially utilize grandmothers to support PrEP uptake and use particularly among pregnant and breastfeeding women. Efforts have been made to enhance the skills of grandmothers to discuss issues related to sex and sexuality with young girls to reduce unintended pregnancy (41) as well as HIV acquisition (42) and their support has been shown to be important for prevention. It is therefore possible that their supportive roles during pregnancy and breastfeeding can be similarly harnessed to promote HIV prevention product uptake and adherence among their daughters and daughters-in-law. Further consultations with grandmothers on how to best engage them, with a focus on PrEP education and adherence support mechanisms and strategies, are needed

LIMITATIONS:

34. Can you say how you mitigated the third and fourth limitations, if at all?

Author’s response: The third limitation could not be mitigated as it was observed during analysis. Attempts were made to mitigate the fourth limitation by ensuring a comfortable and casual discussion space and by introducing the goals of the study and the roles of participants within the framework of the discussion guide (Please see supplied guides attached as supporting information S1, S2 and S3). It was not feasible to facilitate discussions off site in more casual settings due to the need for privacy and audiorecording.

Text has been updated as follows:

Discussion (Lines 560 – 565): Thirdly, it was difficult to ascertain during analysis whether some responses during the discussions were related to mothers or mothers-in-law specifically, as the terminology was interchanged with the more generic “grandmother” term. Fourthly, although facilitators ensured comfortable and casual FGD venues and discussions, responses may have been swayed by social desirability bias due to the FGDs occurring within clinic settings in order to maintain privacy and confidentiality. 

CONCLUSION:

35. Line 438 - Be careful when extrapolating these findings of potentially increased PrEP uptake, to lower rates of HIV acquisition (which also requires adherence to PrEP). Make it clear throughout the paper whether you are speaking about PrEP adherence or uptake.

Author’s response: The framing of the questions in the FGD guides were geared around supporting the general use of PrEP. PrEP uptake or adherence was not specifically stated however data from all group discussions indicated that the decision to use PrEP (PrEP uptake) should be made by pregnant or breastfeeding women alone or together with their partners. Some pregnant and breastfeeding women mentioned that grandmothers could support PrEP use once they are already taking it (PrEP adherence) as did grandmothers. 

Text has been updated as follows: 

Conclusion (Lines 579-583): Although the intensity of these roles differed by setting, these data are indicative that grandmothers’ supportive influence may be extended to support uptake of and adherence to biomedical HIV prevention options and potentially contribute to the decline in HIV acquisition among pregnant and breastfeeding women in these communities. 

36. Line 436 - “With the right framing and approach” - I am curious to hear your suggestions about this approach (maybe in the Discussion, as I mention above).

Author’s response: Text has been updated as follows:

Discussion (Lines 539-550): These data bring to light new insights and opportunities to potentially utilize grandmothers to support PrEP uptake and use particularly among pregnant and breastfeeding women. Efforts have been made to enhance the skills of grandmothers to discuss issues related to sex and sexuality with young girls to reduce unintended pregnancy (41) as well as HIV acquisition (42) and their support has been shown to be important for prevention. It is therefore possible that their supportive roles during pregnancy and breastfeeding can be similarly harnessed to promote HIV prevention product uptake and adherence among their daughters and daughters-in-law. Further consultations with grandmothers on how to best engage them, with a focus on PrEP education and adherence support mechanisms and strategies, are needed.

Sincerely, 

Ms Krishnaveni Reddy

Technical Head: Clinical Trials (Research Centre Clinical Research Site)

Wits Reproductive Health and HIV Institute (Wits RHI)

---

## [Editor Report · Decision Letter 1]

6 Jul 2022

"Ask the way from those who have walked it before" – Grandmothers’ roles in health-related decision making and HIV pre-exposure prophylaxis use among pregnant and breastfeeding women in sub-Saharan Africa

PONE-D-21-27994R1

Dear Dr. Reddy,

We’re pleased to inform you that your manuscript has been judged scientifically suitable for publication and will be formally accepted for publication once it meets all outstanding technical requirements.

Kind regards,

Douglas S. Krakower, MD

Academic Editor

PLOS ONE

Additional Editor Comments (optional):

1. Line 222. Please clarify in the text that it is the study participants, and not the authors, who describe women as behaving irrationally.

2. Line 271. Please remove the word "promiscuity" as it can be stigmatizing, or clarify in the text that this is the term used by participants (and not the authors). 

3. Line 312. Please clarify in this paragraph that the concerns about fetal harms and other negative outcomes are perceived negative outcomes by participants and not based on clinical evidence (to ensure that readers are not misled into thinking these are known negative impacts of PrEP use). Please also do this for Line 336 (i.e. clarify that these are perceptions of harms and not harms that are expected based on clinical evidence or experience), and for the term "bitterness" in Line 350 (as I am not aware of prior descriptions of PrEP as bitter).  

4. Line 394. Please clarify that many of the participants' concerns were not based on clinical evidence, and that PrEP use is generally safe; the text in its current form suggests that the participants' concerns about potential harms from using PrEP are accurate. A more balanced description of the potential harms from PrEP use is needed.

5. In the Limitations section, please also add mention that the study satisfied many but not all of the COREQ checklist items, and give a brief description of why these items were not satisfied. While the authors indicated N/A for some items appropriately (e.g. for repeat interviews), there were also items listed as N/A that could reasonably have been addressed to potentially improve this study, most notably a more detailed description of the researchers, discussion of saturation, and returning transcripts to participants / member checking. 
---

## [Editor Report · Acceptance letter]

23 Aug 2022

PONE-D-21-27994R1 

‘‘Ask the way from those who have walked it before” – Grandmothers’ roles in health-related decision making and HIV pre-exposure prophylaxis use among pregnant and breastfeeding women in sub-Saharan Africa 

Dear Dr. Reddy:

I'm pleased to inform you that your manuscript has been deemed suitable for publication in PLOS ONE. Congratulations! Your manuscript is now with our production department. 

Kind regards, 

on behalf of

Dr. Douglas S. Krakower 

Academic Editor

PLOS ONE